# Rhythm Control in Patients with Heart Failure with Preserved Ejection Fraction: A Meta-Analysis

**DOI:** 10.3390/jcm10184038

**Published:** 2021-09-07

**Authors:** Narut Prasitlumkum, Ronpichai Chokesuwattanaskul, Wisit Cheungpasitporn, Jakrin Kewcharoen, Charat Thongprayoon, Tarun Bathini, Saraschandra Vallabhajosyula, Krit Jongnarangsin

**Affiliations:** 1Division of Cardiology, University of California Riverside, Riverside, CA 92521, USA; 2Division of Cardiology, Department of Medicine, Faculty of Medicine, Chulalongkorn University and King Chulalongkorn Memorial Hospital, Thai Red Cross Society, Bangkok 10330, Thailand; 3Center of Excellence in Arrhythmia Research Chulalongkorn University, Department of Medicine, Faculty of Medicine, Chulalongkorn University, Bangkok 10330, Thailand; 4Department of Internal Medicine, Mayo Clinic, Rochester, MN 55905, USA; charat.thongprayoon@gmail.com; 5Department of Cardiology, Loma Linda University, Loma Linda, CA 92354, USA; jakrinkewcharoen@gmail.com; 6Department of Internal Medicine, University of Arizona, Tucson, AZ 85721, USA; tarunjacobb@gmail.com; 7Section of Cardiovascular Medicine, Department of Medicine, Wake Forest University School of Medicine, Winston-Salem, NC 27101, USA; svallabh@wakehealth.edu; 8Department of Electrophysiology, Division of Cardiovascular Medicine, University of Michigan, Ann Arbor, MI 48109, USA; kritj@med.umich.edu

**Keywords:** rhythm control, HFpEF, atrial fibrillation, heart failure, diastolic heart failure

## Abstract

Background The presence of atrial fibrillation (AF) in patients with heart failure with preserved ejection fraction (HFpEF) dramatically increases higher morbidity and mortality. Recent studies have suggested that early rhythm control may alleviate the burden of poor outcomes. Currently, there remain limited data on whether rhythm or rate control has better efficacy. This study sought to compare both strategies in HFpEF patients with AF. Methods Databases were searched throughout 2020. Studies that reported cardiovascular outcomes amongst HFpEF patients with AF who received either rhythm or rate control were included. Estimates of the effects from the individual studies were extracted and combined using random-effects, a generic inverse variance method of DerSimonian and Laird. Results Five observational studies were included in the analysis, consisting of 16,953 patients, 13.8% of whom were receiving rhythm control. In comparison with rate control, rhythm control was associated with decreased overall mortality rates (pooled RR 0.85, 95% CI 0.75–0.95, with I^2^ = 0%, *p* value = 0.009). Conclusions In HFpEF patients with AF, rhythm control was associated with lower mortality, compared to rate control. Further studies are warranted to validate our observation.

## 1. Introduction

HFpEF has become more prevalent, with an upward trajectory in the older patient population [1]. Recent studies have suggested poor survival rates, at least equivalent to those among HFrEF patients [2,3]. Unfortunately, no current, evidence-based treatments have shown significant mortality improvement with regard to this condition—only improvements to underlying conditions and comorbidities [4,5]. According to a previous study, 65% of HFpEF patients had concomitant AF [6]. It was observed that atrial arrhythmia was correlated with higher mortality rates, HF hospitalization, stroke, and TIA, compared to HFpEF patients with sinus rhythm or other heart failure groups [3,6].

Previous studies failed to show superior clinical benefits from rhythm control to those of rate control. Nevertheless, the most recent trial, EAST-AFNET [7], highlighted the boon of rhythm strategy when it is promptly implemented. Despite this promising finding, more robust clinical data are required to ensure its safety in unhealthy populations, especially HF patients. Currently, there are no current specific schemes for AF treatment in HFpEF patients. In light of the paucity of data and the controversy, we collectively sought to garner, analyze, and summarize systematic reviews and meta-analyses under the hypothesis that rhythm control is associated with improved adverse CV outcomes.

## 2. Methods

### 2.1. Literature Review and Search Strategy

A systematic literature search of MEDLINE, EMBASE, PUBMED, and the Cochrane Database of Systematic Reviews (from the inception of the database to December 2020) was conducted to identify studies comparing cardiovascular outcomes resulting from rate control and rhythm control in HFpEF patients who had a history of AF. The systematic literature review was undertaken independently by two investigators (R.C. and N.P.), applying a search approach provided in online Appendix A, which incorporated the following terms: “Heart failure with preserved ejection fraction”, “HFPEF”, “Diastolic heart failure”, “Rate control”, “Rhythm control”, and “Atrial fibrillation”. No language restriction was applied. A manual search for conceivably relevant studies using references from the included articles was also performed. When there were disagreements between the reviewers, we engaged in discussion until we reached a consensus. This study was conducted according to the PRISMA [8] (Preferred Reporting Items for Systematic Reviews and Meta-Analysis) statement, as described in online Appendix A.

### 2.2. Selection Criteria

Eligible studies included cross-sectional, case–control, or cohort studies that assessed the associations of side effects and safety. They were required to provide the estimated incidence of the effects, prevalence, odds ratio (OR), relative risk (RR), or hazard ratio (HR), with 95% confidence intervals (CI). Inclusion was not limited by study size. Retrieved articles were individually reviewed for their eligibility by the two investigators noted previously. Discrepancies were discussed and resolved by mutual consensus. The Newcastle–Ottawa quality assessment scale (NOS) was used to appraise the quality of a case–control study and the outcome of interest of a cohort study, as shown in Appendix A [9].

### 2.3. Data Abstraction

A structured data collecting form was utilized to derive the following information from each study: the title, the year of the study, the name of the first author, the publication year, the country where the study was conducted, the demographic and characteristic data of the subjects, types of exposure (rate control and rhythm control), and cardiovascular outcomes. For the most accurate analysis, we utilized OR/RR/HR via multivariable adjustment from original studies that provided the available data. Otherwise, we extracted raw data and proceeded with univariate analysis. For comparison, the rate control arm was used as a reference point. Inversion of odds ratios was implemented if rhythm control was designated instead in the original studies [10,11]. The primary outcome in this study was long-term mortality, defined by all-cause mortality at 1 year or more.

### 2.4. Statistical Analysis

Meta-analysis of the combined data to estimate risk ratios (RRs) and 95% CIs was performed using a random-effects, generic inverse variance method of DerSimonian and Laird [12]. The heterogeneity of effect size estimates across studies was quantified using the Q and I^2^ statistics. For the Q statistic, substantial heterogeneity was defined as *p* < 0.10. The I^2^ statistic ranged in value from 0% to 100% (I^2^ < 25%, low heterogeneity; I^2^ = 25% to 50%, moderate heterogeneity; and I^2^ > 50%, substantial heterogeneity) [13]. To assess the influence of each study on overall heterogeneity, a sensitivity analysis was performed by excluding one study each time to determine the overall robustness of the study. Meta-regression was also performed to explore the source of heterogeneity. In accordance with Cochrane, publication bias was assessed using a funnel plot. Funnel plot asymmetry was further confirmed with Egger’s test if there were more than ten studies available [14]. All analyses were performed using STATA version 14.1 (College Station, TX, USA).

## 3. Results

Using our search strategy, a total of 429 potentially eligible articles were identified through Rayyan [15]. One hundred and eighty-three articles were excluded due to duplicated studies. After the exclusion of 226 articles because they clearly did not meet the inclusion criteria, based on article types, methodologies, and outcomes of interest, 22 articles remained for the full-length review. Two were excluded as all participants had undergone catheter ablation. Twelve were excluded because no clear data regarding rate and rhythm control were specified. Two were excluded due to the inclusion of HFrEF patients in the analysis. Lastly, one study was excluded because all participants did not have HFpEF at the beginning of the study. In conclusion, the final analysis included five observational studies (four retrospective cohort studies and one prospective cohort study). The literature retrieval, review, and selection processes are demonstrated in Figure 1. The characteristics and quality assessment of the included studies are presented in Table 1.

### 3.1. Study Characteristics and Quality Assessment

The populations included in our study comprised 16,953 individuals, 13.8% of whom were receiving rhythm control. The included studies’ populations were 64.7% female, with an average age of 71.0 ± 8.0 years old. The median time until follow up was 13.5 months. In the rhythm control group, class III antiarrhythmic agents were primarily used, in 52.3–81.9% of studies. Catheter/surgical ablation was used as a primary method in only 1.0–17.5% of included studies. In the rate control group, beta-blockers were used significantly, in 36.8–89.4% of studies. (Table 2) The NOS rating ranged from 7 to 9, indicating a moderate to high quality among the included studies.

### 3.2. Primary Endpoints Results

With regard to long-term mortality, rhythm control was associated with a lower mortality rate (pooled RR 0.85, 95% CI 0.75–0.95, *p* = 0.009). There was only trivial heterogeneity in this analysis (I^2^ = 0%) (Figure 2).

### 3.3. Exploratory Analysis

To the best of our ability using the available data, we further delved into other possible outcomes from the included studies, including HF admission, CV mortality, and stroke/TIA.

For HF admission, CV mortality, and stroke/TIA, there was no statistical difference between rhythm and rate controls (*p* = 0.116, *p* = 0.597 and 0.613, respectively). However, numerically lower HF admission was observed with the use of rhythm control, as compared to rate control (pooled RR 0.65, 95% CI 0.38–1.10, with I^2^ = 77.2%) (Appendix A).

### 3.4. Sensitivity Analysis and Publication Bias

Publication bias was not found from funnel plots for long-term mortality, CV mortality, HF hospitalization, or stroke/TIA (Appendix A). The sensitivity analysis of the primary outcome to explore heterogeneity showed no significant change in our findings when each study was separately omitted. (Appendix A).

## 4. Discussion

This is the first systematic review and meta-analysis comparing rhythm and rate control treatment strategies in HFpEF patients with concomitant AF. Our analysis suggests that rhythm control was associated with a 15% lower mortality, as compared to rate control. Nevertheless, we did not find any statistical differences between rhythm and rate controls for HF admission rates, stroke/TIA, and CV mortality. Nevertheless, we observed a trend toward lower HF admission rates with the use of rhythm control, compared to rate control.

Counterintuitively, previous landmark studies have not shown clear benefits of rhythm control over rate control, regardless of heart failure status. In 2002, both AFFIRM and RACE studies showed the non-superiority of rhythm control in comparison to rate control [21,22]. Subsequent analysis revealed that negative findings might be primarily driven by poorer drug safety profiles in the rhythm control strategy. The adverse effects potentially mitigate the net clinical benefit that we anticipated from this approach. However, many observations from these two studies warranted an explanation for the null effects, including high crossover rates from rate to rhythm control arms, a similar proportion of patients remaining in sinus rhythm at the end of study, and the early discontinuation of anticoagulation therapy in rhythm groups. Furthermore, both groups had a similar sinus rhythm maintenance rate: about 39%, over the follow-up period. This may partly explain the negative results, supported by the fact that only 39% of patients in the rhythm group were able to maintain sinus rhythm. For this reason, it seems premature to conclude that the rhythm control strategy is an unfavorable approach among AF patients. One real-world study [23] demonstrated the superior efficacy of rhythm control to rate control, statistically lowering CV mortality, stroke/TIA, and HF hospitalization, in contrast to results from RCT, which had stringent inclusion criteria [21,22]. A very recent study, EAST-AFNET, demonstrated better CV outcomes if early rhythm control was implemented [7]. According to these contemporary data, the greater use of this strategy among AF patients in clinical practice shows promise.

The dual presence of HF and AF further jeopardizes cardiovascular morbidity and mortality. This conception is supported by myriad former reports, which have clearly shown greater adverse outcomes in AF patients with HF, as well as in those with HFrEF and HFpEF, compared to patients in sinus rhythm [24,25,26,27,28,29]. Importantly, recent epidemiological data suggested high AF prevalence of up to 65% amongst HFpEF patients, which seems higher than in HFrEF cases, where it is only 53% [6]. Given this significant fact, it is paramount to preemptively screen for and address AF, to prevent its consequences in HF patients. Strategies include risk counseling and proper anticoagulation, as well as rate or rhythm controls.

While many studies have alluded to encouraging endpoints with the use of rhythm control in HFrEF patients who had AF, mainly by catheter ablation [30,31,32], only a few studies [16,17,18,19,20] have focused on HFpEF populations. Despite the limited body of evidence, our meta-analysis suggested potential long-term mortality benefits from rhythm control. Considering our included studies as a whole, the largest proportion of participants was from Kelly et al. [16]. Although it may differ in the overall interpretation, our sensitivity analysis demonstrated a similar long-term mortality trend, affirming our study’s robustness. Interestingly, other studies [17,18,19,20] showed only a trend to decreased long-term mortality, while Kelly et al. [16] demonstrated a statistical association with rhythm control. We believe the AF duration, numbers of participants, use of anticoagulation, direct oral anticoagulation vs. warfarin, and differences in therapy modes (medications, ablation, or cardioversion), play crucial roles in each study’s findings. Unfortunately, none of these factors were reported, which precluded further sub-analyses from characterizing their implications. In this regard, our findings should only be considered hypothesis-generating, although positive outcomes were observed in this study.

Physiologically, an irregular atrial contraction would result in dramatically decreased preload and increased LV filling pressure [33,34], compromising hemodynamics in the case of HFpEF. Loss of atrial kick can lead to decompensated HF, as left ventricular filling relies more on atrial contraction to compensate for a decline in left ventricular compliance. Despite appropriate diuresis in these patients, intractable tachycardia may ensue, as the reflex sympathetic tone is activated in response to intravascular depletion, which further perpetuates dysrhythmias [35]. To address this abnormal response, a rhythm control plan is reasonable, owing to favorable outcomes in many studies [23,30,36]. Contemporarily, the use of catheter ablation has become widespread due to higher efficacy in restoring sinus rhythm, as well as reduced CV risks [37,38]. The mortality improvement noted in our study may partly derive from this possible mechanism. Nevertheless, more studies are warranted to determine other possible explanations for mortality advantages.

While a reduction in all-cause mortality was noted, our study did not find a statistical association between rhythm control and CV mortality, stroke/TIA, and HF admission. These outcomes were not originally specified, but rather resulted from exploratory analyses to investigate the current data using real-world experience. Of note, these outcomes originally resulted from subgroup analyses of the included studies, which were subject to residual biases and underpowered due to limited numbers of participants. In our opinion, the current evidence is not enough to firmly reject rhythm control, owing to the negative results. Therefore, these findings should be regarded as hypothesis-generating rather than confirmatory.

Whether the mortality benefit in our analysis is driven by a reduction in CV or non-CV mortality remains unclear. Irretrievable information from each included study [16,17,18,19], required to assess the non-CV mortality benefit, unfortunately prevented our sub-analysis from exploring this perspective. Given the scarce data in AF management for HFpEF patients, dedicated randomized clinical trials are required to investigate the best strategy between rate and rhythm control in AF patients with HFpEF.

Despite statistically non-significant results, we observed a trend of lower HF admission in the rhythm control group. In a sense, restoring sinus rhythm should intuitively improve cardiac performance and prevent adverse CV sequelae. In one study [39], successful conversion from AF to sinus rhythm conferred substantial exercise tolerance, higher peak oxygen consumption, and an improvement in LVEF, despite already being within the normal LVEF range. Another study demonstrated an improvement in cardiac geometry dimensions in both systolic and diastolic metrics [40]. In the same study, maintaining sinus rhythm also ameliorated LV filling pressure, which subsequently reduced LA pressure. In theory, the incidence of heart failure exacerbation should have been reduced, owing to the attenuation of HFpEF pathophysiology, as previously discussed. Nonetheless, our study did not reach a sufficient point of statistical significance to affirm this hypothesis. In addition to our study possibly being underpowered, we believe a shorter follow-up period in the study of Kelly et al. study, compared to the other two studies, potentially explains the non-statistical findings in our analysis. Perhaps longer observation might be required to restore diastolic function in patients with AF to achieve the HF benefit after receiving rhythm control.

## 5. Limitations

Our study had several limitations. First, owing to its observational nature, residual biases cannot be completely excluded. Differences in methodologies, demographics, and backgrounds still carry inevitably unrecognized confounders. In addition, it is possible that healthier populations were selectively chosen for the rhythm control approach, considering the side effects of antiarrhythmic agents. In this regard, the random-effects model and sensitivity analyses were employed, confirming the study’s robustness [41]. Second, only small populations were included in this study. We acknowledge that this is our major limitation, especially with regard to our exploratory analyses, for which only scant numbers of studies were discovered. More vigorous information is required, especially randomized control trials, to prove our preliminary analyses. Despite this fact, this is the most up-to-date systematic review and meta-analysis, providing new insights and hypothesis-generating viewpoints for HFpEF management. Third, our study was not perfectly adjusted for several crucial factors, such as AF burden, right ventricular function, pulmonary artery pressures, and other relevant echocardiographic findings, primarily due to insufficient data. Nevertheless, our analysis utilized the available OR/RR/HR with multivariable adjustments where possible, minimizing the confounding effects from unrecognized factors. Fourth, despite including HFpEF patients exclusively, certain cardiomyopathies, such as hypertrophic cardiomyopathy and amyloidosis, were not specified in the included studies, precluding further analysis stratified by cardiomyopathy type. Further studies should focus on these subtypes. Fifth, the use of AF ablation in our study seems diminutive. As a result, we were unable to further assess the true benefit of individual rhythm control methods. In spite of this, our main findings seem promising, supporting a greater use of rhythm control in the current era, as well as warranting more studies to evaluate the efficacy of AF ablation in the HFpEF population.

## 6. Conclusions

This is the first systematic review and meta-analysis showing the potential benefits of rhythm control compared to rate control in HFpEF patients with AF. Our study demonstrated that the employment of a rhythm control strategy was associated with a reduction in long-term mortality of up to 15%. Rhythm control, however, was not associated with HF admission rate, CV mortality, or stroke/TIA outcomes. Nevertheless, there was a numerical trend toward 40% lower HF hospitalization in the rhythm control arm. We are optimistic that further studies may be conducted in the future to validate our hypothesis-generating results.

## Figures and Tables

**Figure 1 jcm-10-04038-f001:**
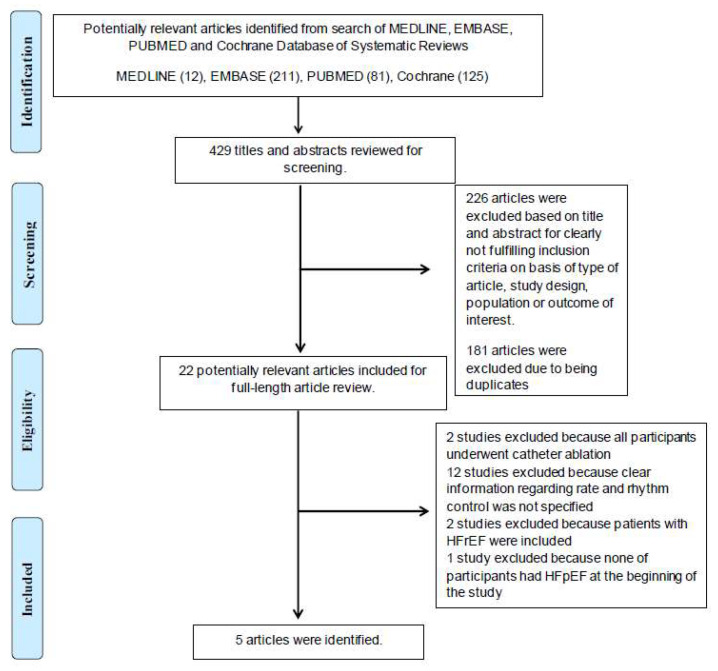
The literature retrieval, review, and selection processes.

**Figure 2 jcm-10-04038-f002:**
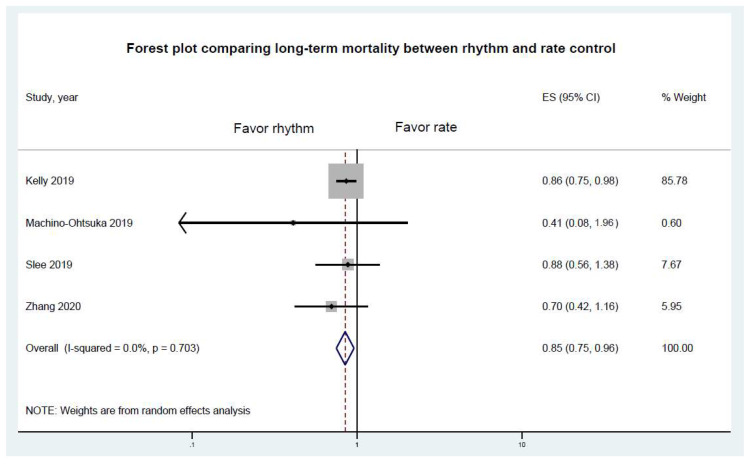
Forest plot comparing risk of long-term mortality between rhythm and rate controls. Horizontal lines represent the 95% Cis, with marker size reflecting the statistical weight of the study, using the random-effects model. A diamond data marker represents the overall adjusted OR and 95% CI for the outcome of interest.

**Table 1 jcm-10-04038-t001:** Characteristics of included studies.

Study	Kelly 2019 [16]	Machino–Ohtsuka 2019 [17]	Slee 2019 [18]	Zhang 2020 [19]	Zhirov 2019 [20]
Types	Retro	Retro	Retro	Retro	Pros
Country	USA	Japan	USA	USA	Russia
Participants	HF patients >65 years old with concurrent AF who were discharged alive	HFpEF patients >20 years old with AF	>65 years old HF patients with prior AF who were at high risk of stroke	HF patients >18 years old with prior or current AF	HF patients >18 years old with prior or current non-valvular AF
Database	Medicare data from 2008 to 2014	Multicenter from 2012 to 2015	AFFIRM registry	REP from 2000 to 2014	Multicenter from 2015 to 2016
Exclusion	Patients who did not receive either rhythm or rate control strategies, patients who were not admitted	Younger than 20 years old, prior MI, valvular disease requiring intervention, history of pacemaker implantation, severe lung and liver disease	N/A	Patients without documented EF, patients with AF who died within 1 year	Recent stroke/TIA, recent MI, valvular AF, BiV implantation, severe life-limiting comorbidities, recent VTE
HFpEF criteria	HF with EF > 50%	HF with EF > 50%	HF with EF > 40%	HF with EF > 50%	HF with EF > 50%
Mean EF (%)	58.0	63.8 ± 8	N/A	61.2 ± 6.7	60.0 ± 5.0
Mean age (years)	83.0	71 ± 8	70.9 ± 8.7	79.2 ± 11.1	72.0
Sex (Female%)	65.8	39.9	40.1	60.2	65.4
CHA2DS2-VASc	N/A	Rate: 4.2 ± 1.4, Rhythm: 3.9 ± 1.3	N/A	Rate: 5.2 ± 1.7, Rhythm: 4.7 ± 2	N/A
Total participants	15,682	283	349	447	387
Hypertension	Rate: 81.9%, Rhythm: 83.8%	Rate: 75.0%, Rhythm: 80.4%	Rate: 80.1%, Rhythm: 76.4%	Rate: 85.4%, Rhythm: 75.0%	68%
Diabetes	Rate: 36.2%, Rhythm: 36.0%	Rate: 31.3%, Rhythm: 34.6%	Rate: 33.3%, Rhythm: 20.8%	Rate: 36.8%, Rhythm: 30.0%	23%
Coronary artery disease	Rate: 45.8%, Rhythm: 48.7%	Rate: 17.0%, Rhythm: 25.2%	Rate: 17.0%, Rhythm: 10.7%	Rate: 16.5%, Rhythm: 15.0%	70%
CVA/TIA	Rate: 19.3%, Rhythm: 17.5%	Rate: 9.1%, Rhythm: 12.1%	Rate: 11.7%, Rhythm: 16.9%	Rate: 22.6%, Rhythm: 2.5%	15%
Proportion of paroxysmal AF	N/A	37.4%	N/A	N/A	37.2%
AF duration (years)	N/A	5.8 ± 6.7	N/A	4.2 (IQR 2–9)	N/A
Mean follow up (months)	12	24	48	49.2	12
Proportion of rhythm control	11.8%	37.8%	51.0%	15.9%	40.6%
Proportion of rate control	78.2%	62.2%	49.0%	74.1%	59.4%
Adjusted variables	Age, sex, race, prior MI, hypertension, hyperlipidemia, smoking history, prior CVA/TIA, DM, CKD, anemia, PVD, prior HF, COPD	Age, sex, body mass index, vital signs, prior MI, hypertension, hyperlipidemia, CKD, DM, medications, laboratory data, LVEF, LA volume, E/E’, TRPG, GLS, LV mass	Age, sex, failed antiarrhythmic drugs, hypertension, MI, stroke, DM, hypertension, cardiomyopathy, valvular heart disease	Age, sex, time interval from HF to AF, body mass index, hypertension, COPD, prior MI, prior stroke	N/A

**Table 2 jcm-10-04038-t002:** Rate and rhythm control modalities.

Study	Rhythm Control	Rate Control
Kelly 2019 [16]	Class III 80.9%, Cardioversion 13.6%, AF ablation 1%, Unspecified 11.4%	Beta blocker 89.4%, CCB 25.3%, Digoxin 17.1%
Machino–Ohtsuka 2019 [17]	Class IA 6.5%, Class IC 37.4%, Class III 52.3%, Unspecified 10.3%	Beta blocker 54.5%, CCB 42.6%
Slee 2019 [18]	Class IC 3.9%, Class III 78.7%, Unspecified 8.4%	Beta blocker 36.8%, CCB 39.2%, Digoxin 51.9%
Zhang 2020 [19]	N/A	Beta blocker 80%, CCB 30%, Digoxin 20%
Zhirov 2019 [20]	N/A	N/A

Abbreviations: AF: atrial fibrillation; AFFIRM: The Atrial Fibrillation Follow-up Investigation of Rhythm Management; BiV: biventricular ventricular resynchronization; CCB: calcium channel blocker; CKD: chronic kidney disease; COPD: chronic obstructive pulmonary disease; CVA: cerebrovascular accident; DM: diabetes mellitus; EF: ejection fraction; GLS: global longitudinal strain; HF: heart failure; HFpEF: heart failure with preserved ejection fraction; LA: left atrium; LV: left ventricle; MI: myocardial infarction; N/A: not applicable; PVD: peripheral vascular disease; REP: Rochester Epidemiology Project; TIA: transient ischemic stroke; TRPG: tricuspid regurgitation peak gradient; VTE: venous thromboembolism.

## Data Availability

The data for this systematic review and all potentially eligible studies are publicly available through the Open Science Framework (URL: https://osf.io/mn2eu/).

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
