# Peer review of "Rhythm Control in Patients with Heart Failure with Preserved Ejection Fraction: A Meta-Analysis"

_jcm, 2021, doi:10.3390/jcm10184038_

Round 1

Reviewer 1 Report

In the current manuscript, Prasitlumkum and colleagues investigated the association of rhythm and rate control with cardiovascular outcomes in an atrial fibrillation population with heart failure with preserved ejection fraction. The methodology is well described. After screening 429, they included 5 articles in their analyses. They found a 15% lower overall mortality associated with rhythm control, but no significant associations for the other outcomes. The authors should be congratulated on their analyses with regards to the relevance of their research question.

Major comments

  1. Line 175-177: The authors should lower the tone on possible causality in their discussion. This cannot be inferred from their current data.
  2. Why do the authors think that a significant association with overall mortality but not CV mortality was observed?
  3. The authors should discuss the very likely residual confounding regarding outcomes. It is conceivable, that healthier patients in observational studies are more likely to receive rhythm control.  
  4. Can the authors clarify, how they arrive at the estimates given in their figures? Are these the original estimates from the articles or are these adjusted estimates within their analyses? If these were original estimates, I cannot find them as given in the original papers as for example in Figure 3.

Minor comments

  • Table 1: The Table should be better formatted for rate and rhythm control modalities.
  • In the Methods they authors write that they "sequentially and cumulatively excluded studies that accounted for the largest share of heterogeneity until I2 was less than 50%". How many studies were excluded due to this criteria?
  • Figure 3 says "Zhirov 2020" which should be "2019"
  • What do the authors think about the informative value of funnel plots including only 2-3 studies?
  • The whole manuscript should be spell-checked and revised regarding english language.
  • The authors should use consistently one decimal for numbers.

Author Response

Reviewer 1

In the current manuscript, Prasitlumkum and colleagues investigated the association of rhythm and rate control with cardiovascular outcomes in an atrial fibrillation population with heart failure with preserved ejection fraction. The methodology is well described. After screening 429, they included 5 articles in their analyses. They found a 15% lower overall mortality associated with rhythm control, but no significant associations for the other outcomes. The authors should be congratulated on their analyses with regards to the relevance of their research question.

Response: Appreciate the reviewer’s comment. We hope our study provides more attention to AF management in HFpEF patients. We are open to any comments or suggestions to improve our manuscript quality and readability. 

 Major comments

  1. Line 175-177: The authors should lower the tone on possible causality in their discussion. This cannot be inferred from their current data.

Response: Appreciate the reviewer’s recommendation. We have toned down on our synopsis statement on the first paragraph of our discussion from “This is the first systematic review and meta-analysis comparing rhythm and rate control treatment strategy in HFpEF patients with concomitant AF. Our analysis suggested rhythm control may confer long-term mortality benefit, a 15% reduction in mortality, compared to rate control.” to “Our analysis suggested rhythm control in AF patients with HFpEF was associated with a 15% lower mortality, compared to rate control.”

  1. Why do the authors think that a significant association with overall mortality but not CV mortality was observed?

Response: Appreciate the reviewer’s recommendation. We think this is mainly driven by the underpower owing to very few included studies are with available information to fully analyze. Also, data deriving from available studies were also conducted in a secondary analysis manner which precludes the best analysis. Due to this limitation, we have revised our discussion highlighted in red. We extracted the new paragraph dedicated to this issue as follows:

“While all-cause mortality reduction was noted, our study did not find statistical association between rhythm control and CV mortality, stroke/TIA and HF admission. However, there is a trend toward lowering hospitalization in rhythm control with marginal 95% CI that almost achieves statistical significant. These outcomes were originally non-prespecified but rather exploratory analyses to investigate the current data in real-world experience. Of note, these outcomes were originally conducted as subgroup analyses from the included studies, which were subjected to residual biases and underpowered due to limited numbers of participants. In our opinion, current evidence is not enough to firmly negate the rhythm control owing to the negative results. These findings should be regarded as a hypothesis-generating rather than confirmatory viewpoints.”

  1. The authors should discuss the very likely residual confounding regarding outcomes. It is conceivable, that healthier patients in observational studies are more likely to receive rhythm control.  

Response: Appreciate reviewer’s recommendation. This is very crucial point that may have confounded our overall results in addition to possible underpower of study, which we described in the prior reviewer’s response. Given this limitation, we have further discussed this issue in our limitation part as following

“First, owing to observational nature, residual biases cannot be completely excluded. Despite low heterogeneity, differences in methodologies, demographics, and backgrounds still carry inevitably unrecognized confounders. In addition, it is possible that healthier populations were selectively opted for rhythm control approach considering potential side effects from antiarrhythmics agents. Owing to this reason, random-effects model, subgroup analyses, and sensitivity analyses were all employed for appropriate adjustment confirming study’s robustness.”

  1. Can the authors clarify, how they arrive at the estimates given in their figures? Are these the original estimates from the articles or are these adjusted estimates within their analyses? If these were original estimates, I cannot find them as given in the original papers as for example, in Figure 3.

Response: Appreciate reviewer’s suggestion. We do apologize for the inconsistency in some estimates provided in our figures. This is due to a technical issue in which we have re-edited all funnel plots as suggested.

To clarify, all estimates derive from original studies. As we described in our methodologies, we opted for multivariable adjusted estimates first. Otherwise, univariate estimates were chosen. If both criteria were not met, available raw data provided in the original study were extracted and estimated accordingly. The extracted statements mentioned in the method section are as following

“A structured data collecting form was utilized to derive the following information from each study, including title, year of the study, name of the first author, publication year, country where the study was conducted, demographic and characteristic data of subjects, types of exposures (rate control, rhythm control) and cardiovascular outcomes. For the most accurate analysis, we utilized OR/RR/HR from multivariable adjustment from original studies that provided the available data. Otherwise, we extracted raw data and proceeded with univariate analysis.”

For Zhirov estimation regarding HF admission, inversion of odds ratios was employed as in the original study, and it was represented using rhythm control as a reference, while our study purposely used rate control as our referent point. This strategy was previously exerted in our previous publications (PMID: 33633044, 30650054)

Minor comments

  • Table 1: The Table should be better formatted for rate and rhythm control modalities.

Response: Appreciate reviewer’s recommendation. To improve readability and data representation, dedicated table 2 was formatted for rate and rhythm control information.

  • In the Methods, the authors write that they "sequentially and cumulatively excluded studies that accounted for the largest share of heterogeneity until I2 was less than 50%". How many studies were excluded due to these criteria?

Response: Appreciate reviewer’ recommendation. We apologized for the misunderstanding. We did not exclude any studies until I2 became less than 50% but only one each time for our sensitivity analysis. We have rephrased this sentence from the aforementioned to “To assess the influence of each study on overall heterogeneity, a sensitivity analysis was performed by excluding one study each time to assess the overall study robustness. “

  • Figure 3 says "Zhirov 2020" which should be "2019"

Response: Appreciate reviewer’s recommendation. It has been revised to 2019

  • What do the authors think about the informative value of funnel plots, including only 2-3 studies?

Response: appreciate the reviewer’s recommendation. Thank you for pinpointing this issue. Funnel plots with 2-3 studies were all from exploratory analyses, which seems less meaningful than the primary outcome. Hence, these figures were removed from the main manuscript but kept in our supplementary file instead. Information regarding exploratory analyses seems already sufficiently narrated in the main manuscript.

  • The whole manuscript should be spell-checked and revised regarding English language.

Response: Appreciate reviewer’s recommendation. We have native English speakers proofread, spell-checked and extensively revised the whole manuscript as suggested.

  • The authors should use consistently one decimal for numbers.

Response: Appreciate reviewer’s recommendation. We have changed all to one decimal for numbers. We nevertheless chose to preserve two decimals for all statistical annotations as this method should better represent our data.

We greatly appreciated the editor and reviewer’s time and comments to improve our manuscript.

Reviewer 2 Report

This article is a meta-analysis comparing rhythm control and rate control in patients with HFpEF. It is a well-written article with appropriate methology, on an interesting topic.

I have only one issue to comment.
For HF admission, rhythm control group showed numerically lower HF admission compated to rate control, but it was NOT significant (p=0.081).  However, the authors mention the difference in these numbers several times in the Results and Discussion section of the paper, giving readers the nuance that there is a near-statistical difference.
However, strictly speaking, it was not statistically significant, and it would be misleading for readers to assume that actual trends were suggested, or supported. Therefore, it is necessary to emphasize that there was no statistical significance in this meta-analysis.

Author Response

Reviewer 2

This article is a meta-analysis comparing rhythm control and rate control in patients with HFpEF. It is a well-written article with appropriate methodology on an interesting topic.

I have only one issue to comment.

For HF admission, rhythm control group showed numerically lower HF admission compared to rate control, but it was NOT significant (p=0.081).  However, the authors mention the difference in these numbers several times in the Results and Discussion section of the paper, giving readers the nuance that there is a near-statistical difference.

However, strictly speaking, it was not statistically significant, and it would be misleading for readers to assume that actual trends were suggested or supported. Therefore, it is necessary to emphasize that there was no statistical significance in this meta-analysis.

Response: appreciate reviewer’s recommendation. Firstly, we do apologize reviewer for p value in HF readmission. Due to technical error, the correct p value, which is 0.116, was not stated, but instead, 0.81 was mistakenly populated. Therefore, we have proofread and revised all manuscripts again to ensure the accuracy.

In agreement with the reviewer, p value > 0.05 should be considered non-statistical significance. To avoid biasing the readers, we have rephrased deviating statements in our result and discussion parts as well as removed the misleading statement in our abstract.

We greatly appreciated the editor and reviewer’s time and comments to improve our manuscript.

Round 2

Reviewer 1 Report

The authors addressed my comments sufficiently. Congratulations on this study!